# The Influence Mechanism of a Scanning Strategy on the Fatigue Life of SLM 316L Stainless Steel Forming Parts

**DOI:** 10.3390/ma18194571

**Published:** 2025-10-01

**Authors:** Huijun Ma, Xiaoling Yan, Huiwen Fu

**Affiliations:** School of Computer and Artificial Intelligence, Beijing Technology and Business University, Beijing 102488, China; mahuijun@th.btbu.edu.cn (H.M.);

**Keywords:** SLM forming parts, additive manufacturing, scanning strategies, fatigue life

## Abstract

The quality of SLM formed parts is one of the key factors of the promotion and application of additive manufacturing technology. The scanning strategy fundamentally affects the fatigue life of SLM 316L stainless steel parts by regulating residual stress, defect distribution, and microstructure. Three different scanning strategies (meander scanning, stripe scanning, and chessboard scanning) were adopted to prepare the specimens. High cycle fatigue loading was applied to SLM 316L stainless steel specimens prepared by different scanning strategies. The thermal conductivity characteristics during the SLM part forming process were analyzed based on scanning electron microscopy observations of microstructure of SLM specimens, and the mechanism of residual stress and internal defect generation were revealed. The scanning direction determines the growth direction of the grains, thereby affecting the anisotropy and overall fatigue performance of SLM 316L stainless steel parts. The scanning path determines the overlap and lap joint of the melt pool, directly affecting the number, size, and location of pores and incomplete fusion defects. The scanning strategy affects the distribution and magnitude of residual stresses by changing the path of heat source movement. Theoretical analysis and experimental verification results indicate that the selection of a scanning strategy is an effective method for optimizing the fatigue performance of SLM parts.

## 1. Introduction

Selective laser melting (SLM) [1,2,3] is one of many important technologies in additive manufacturing. When a 316L stainless steel part is formed by SLM technology, the discrete contour information of the three-dimensional model is used to control a high-energy laser beam to melt metal powder layer by layer, and then multiple layers of undulating planes accumulate in the height direction of the formed part to form a three-dimensional part. SLM technology has shown strong advantages in the manufacturing of complex shaped components, but there are many factors that affect specimen quality during the forming process, such as laser power, scanning speed, scanning strategy, and rotation angle [4,5,6,7,8]. Among these influencing factors, scanning strategy is one of the key factors affecting SLM formed parts, which directly affects the density, residual stress, internal defects, and surface quality of the formed parts. The presence of residual stress and internal defects can greatly reduce the static strength and fatigue strength of formed parts, and shorten the fatigue life of components.

At present, the main research methods both domestically and internationally are targeted at specific forming materials, using different process parameters, preparing a large number of specimens through orthogonal experiments or single-factor experiments, measuring the fatigue life of the specimens, and determining the optimal process parameters [9,10,11,12,13]. Qian [14] conducted in situ tests on fatigue specimens of Ti-6Al-4V prepared by selective laser melting at different temperatures (25 °C, 200 °C, 400 °C, 600 °C) in horizontal and vertical printing directions. The results showed that anisotropy caused by the printing direction led to fatigue property differences. Brandl [15] investigated the fatigue properties of AlSi10Mg samples prepared by selective laser melting by conducting high-cycle fatigue tests on 91 samples manufactured in different build directions (0°, 45°, 90°) under different temperatures (30 °C, 300 °C). The results indicated that post-heat treatment had the greatest impact on fatigue resistance, while the build direction had the least effect. Although the aforementioned studies have explored the factors influencing the fatigue resistance of samples prepared by selective laser melting (SLM) from different perspectives, objectively described the phenomenon of fatigue damage, and conducted statistical analysis of data through experiments, they lack in-depth theoretical analysis, which confines the research to a relatively superficial level. Theoretical research and practical experience have shown that fatigue damage in SLM formed parts is the cumulative effect of residual stresses and defects in stacked micro units at various scales. Therefore, solving fundamental scientific problems, such as how non-steady state multiple factors affect the solidification shrinkage stress of the selected laser melting “micro melt pool”, the stress of the three-dimensional accumulation body from small to large, and how defects evolve, is the key to revealing the fatigue life evolution law of SLM formed parts [16,17,18,19,20,21,22].

The scanning strategy greatly affects the heat conduction and stress accumulation during the SLM workpiece forming process [23,24,25,26]. Fatigue failure is a process involving crack initiation → crack propagation → final fracture [27,28,29,30]. Fatigue cracks almost always initiate at stress concentration points on or near the surface, such as pores, lack-of-fusion defects, or valleys of surface roughness. Tensile residual stresses on the surface directly increase the local stress level at these defects, facilitating microplastic deformation and thereby significantly accelerating the crack initiation process. Cracks open and propagate under tensile stress. Tensile residual stress increases the stress intensity factor range (ΔK) at the crack tip, which increases the amount of crack growth per load cycle and accelerates the crack propagation rate. Thus, tensile residual stresses significantly degrade the fatigue performance of SLM components. Conversely, compressive residual stresses on the surface act like protective “armor,” counteracting part of the applied tensile stress and reducing the actual stress amplitude at stress concentration points. This delays or even prevents the initiation of micro-cracks. Furthermore, compressive residual stress at the crack tip makes it difficult for the crack to open during loading, effectively reducing the stress intensity factor range (ΔK). This slows down or even halts crack propagation. In some cases, if the compressive residual stress is sufficiently high, it can completely suppress the growth of small cracks. Therefore, compressive residual stresses significantly enhance the fatigue performance of SLM components.

In meander scanning strategy, the prolonged unidirectional movement of the laser leads to intense heat accumulation along one direction, generating extreme temperature gradients. These significant temperature gradients are the primary cause of high thermal stress and subsequent high residual tensile stresses. Grains grow epitaxially along the direction of the highest heat flow (generally the scanning direction), forming coarse columnar crystals that extend across multiple deposited layers. The extremely high residual tensile stresses significantly reduce fatigue strength and promote early crack initiation and rapid propagation. The coarse columnar grain structure also provides an easier path for crack growth. Components may even crack or deform during the printing process due to excessive stress. In the stripe scanning strategy, each layer is divided into multiple narrow strips. The laser performs linear scanning within each strip, but the scanning direction of adjacent strips is typically rotated by a certain angle (e.g., 67°). The shorter scanning path limits heat accumulation in any single direction. Frequent turns and direction changes interrupt long-range heat flow, resulting in a more uniform heat distribution and reduced temperature gradients. Due to the frequent changes in scanning direction, the growth direction of grains is altered in different areas, breaking the continuous penetration of coarse columnar crystals and forming a more equiaxed and finer grain structure. The combined effect of lower residual stress and a finer, more isotropic microstructure effectively hinders crack initiation and propagation, significantly improving fatigue life. In the chessboard scanning strategy, each layer is divided into many small squares (e.g., 5 mm × 5 mm “islands”). The laser scans these islands in a random or specific sequence, and the scanning direction within each island can vary. This disperses the heat input across the entire printing area, with each small island acting as an independent melting and solidification unit, allowing sufficient time for adjacent solidified areas to cool. This approach maximizes the dispersion of thermal stress and avoids the accumulation of long-range stress. Each island forms its own columnar crystal zone, but due to the small size of the islands and the random scanning directions, the overall grain structure becomes finer and less anisotropic. The extremely low macro residual stress means components have a lower risk of deformation, providing a good foundation for high-fatigue life. However, if the island boundaries are not properly handled, introduced micro-defects may offset the benefits of low residual stress and instead become the origin of fatigue cracks, leading to reduced fatigue life. Therefore, optimizing the island size and scanning sequence is crucial.

Choosing an appropriate scanning strategy can significantly reduce the generation and accumulation of residual stresses and defects inside the workpiece, improve the forming quality of the workpiece, and extend the fatigue life [31,32,33]. Testing was conducted to detect residual stresses in SLM 316L stainless steel specimens prepared by different scanning strategies. This included observing the internal microstructure of the specimen through scanning electron microscopy, analyzing the thermal conductivity characteristics during the SLM workpiece forming process, and revealing the mechanism of residual stress and internal defect generation. The accumulation of residual stress and defects will inevitably affect the fatigue life of SLM specimens. The verification of high-cycle-fatigue loading experiments provides a basis for selecting appropriate scanning strategies for SLM workpieces with different working intensity requirements, reducing the significant losses caused by insufficient fatigue life of SLM workpieces, and striving for greater space for the development prospects of SLM technology.

## 2. Experimental Procedures

### 2.1. Material and Specimen Preparation

316L stainless steel spherical metal powder was used in the experiment. The chemical composition of the metal powder is shown in Table 1. A scanning electron microscope image of the powder is shown in Figure 1, with an apparent density of 4.39 g/cm^3^. Figure 2 shows a particle size distribution diagram of 316L stainless steel powder.

The AM400 (Renishaw, Gloucestershire, UK) selective laser melting additive manufacturing system was used. The system is equipped with a safe change filter, optical control software (QuantAM 2021), and an airflow and window protection system, and provides a 400 W optical system with a laser beam diameter of 70 μm. The maximum size of the formed specimen -is 250 mm × 250 mm × 300 mm. During the specimen preparation process, inert gas nitrogen will be filled into the working environment to ensure the stability of the working environment.

There are many factors that affect the fatigue life of the specimen during SLM molding, such as laser power, scanning speed, scanning strategy, and rotation angle. Among these influencing factors, the scanning strategy is one of the key factors affecting the fatigue life of SLM molded parts. Meander, stripe, and chessboard are three typical scanning strategies used in the AM400 SLM manufacturing system. The meander scanning strategy employs continuous long-line scanning, characterized by a simple path generation algorithm, uninterrupted scanning process, and ease of equipment control, which typically results in high scanning efficiency. It is suitable for simple components. The stripe scanning strategy divides long lines into medium-length segments and alternates their scanning directions. As one of the most mainstream and widely used strategies in industrial applications, it strikes an excellent balance between controlling stress and maintaining efficiency, making it applicable to most structural parts. The chessboard scanning strategy partitions the layer into small islands, maximizing the dispersion of the heat source. It is used in scenarios requiring extremely high control of residual stress, such as aerospace applications, including engine casings and large support structures. The aforementioned three scanning strategies have well-defined applications and encompass a wide range of part types, which is why they were adopted in this study to fabricate SLM specimens. Modern advanced SLM equipment often offers more complex hybrid scanning strategies, and the research findings on these three scanning strategies can provide robust support for related studies.

This study focuses on the impact of scanning strategies on the fatigue life of SLM parts. Apart from the scanning strategies, the preparation and testing of the specimens were conducted under completely identical conditions. Therefore, the influence of specimen preparation procedures does not compromise the reliability of the research findings. Three different scanning strategies (meander scanning, stripe scanning, and chessboard scanning) were used to prepare the specimens, and the main processing parameters used in the experiment are shown in Table 2. The build orientation is 0°. The size of the specimen is shown in Figure 3 in mm (millimeters). The scanning paths and sequences of three scanning strategies are shown in Figure 4. During the forming process, a partitioned rotation scanning mechanism was adopted, and the scanning line deviation angle between adjacent layers located in the same area is 67°. Figure 4a shows the scanning path of the Nth layer, and Figure 4b shows the (N + 1)th layer scanning path. The numbers in Figure 4 represent the sequence of laser scanning areas. After the preparation was completed, the specimens were polished with waterproof abrasive paper of 200 to 2000 meshes, and the surfaces of the specimens were wiped.

The path of the meander scanning strategy is relatively simple. The scanning path in the same section is a set of parallel lines, and the spacing between adjacent scanning lines is smaller than the diameter of the laser spot. The specimen can be formed faster and more efficiently by the meander scanning strategy. However, due to the long scanning line and large heat accumulation, significant residual stress will be generated, making it unsuitable for forming high-quality and large components.

When the stripe scanning strategy is adopted, the scanning area is divided into several elongated regions which are separated by equidistant parallel lines. The scanning lines are kept perpendicular to the parallel lines and each region is scanned one by one. Compared with the meander scanning strategy, when the specimen is formed by the stripe scanning strategy, the scanning line is shorter and the accumulated heat is lower, resulting in higher density.

When the chessboard scanning strategy is adopted, the required cross-section is divided into several square regions, with a small amount of overlap allowed between adjacent square regions. The scanning directions between adjacent regions are perpendicular to each other, so that each layer has two scanning vectors perpendicular to each other, reducing the anisotropy inside the SLM formed specimen. Firstly, the square areas with vertical scanning vectors are scanned continuously, and then the square area with horizontal scanning vectors are scanned. The length of the scanning line is equal to the length of the square edge. There is a small amount of overlap between adjacent square areas, and the density of SLM formed specimens is greatly improved.

### 2.2. Tensile Experiment

When conducting fatigue experiments on SLM 316L stainless steel specimens, it is necessary to determine the maximum loading stress. Therefore, before conducting fatigue experiments, it is necessary to stretch the SLM 316L stainless steel specimens to obtain parameters such as the yield strength and maximum strain of the material, providing reference for subsequent experiments. A total of fifteen specimens were prepared, with five specimens for each scanning strategy (meander, stripe, and chessboard scanning strategies). The stress–strain experiment was conducted on a WDW-200E tensile testing machine. Under the same conditions, tensile experiments were conducted on SLM 316L stainless steel specimens prepared by three different scanning strategies. The tensile property parameters of the specimens are shown in Table 3 (the specimens are marked as M, S, and C according to meander scanning, stripe scanning, and chessboard scanning). The experiment results show that the tensile properties of SLM specimens vary with the scanning strategy. In the best tensile performance of the specimens formed by chessboard scanning, due to the use of partitioning and a staggered scanning sequence, the scanning line is short, the preheating between adjacent melt channels is sufficient, the temperature gradient is small, and the accumulation of residual stress is reduced.

### 2.3. Fatigue Loading Experiment

High cycle fatigue loading was applied to SLM 316L stainless steel specimens. Fatigue testing equipment is shown in Figure 5. According to the tensile test results of SLM 316L stainless steel specimens, the maximum stress in the fatigue experiment was set to 400 MPa (400 MPa < 461 MPa), the stress ratio was set to 0.1, and the fatigue loading frequency was 10 Hz. The fatigue load is shown in Figure 6.

## 3. Experimental Results and Analysis

### 3.1. Experimental Results

A total of fifteen specimens were prepared, with five specimens for each scanning strategy (meander, stripe, and chessboard scanning strategies). Under identical experimental conditions, SLM 316L stainless steel specimens were subjected to high cycle fatigue loading until the specimens fractured. The specimens are marked as group M, S, or C according to meander scanning, stripe scanning, and chessboard scanning. The fatigue loading results are shown in Figure 7. As can be seen, the fatigue cycle loading times of the SLM 316L stainless steel specimens formed by the meander, stripe, and chessboard scanning strategies are 60,492 ± 285, 64,390 ± 291, and 68,530 ± 283, respectively. The Kruskal–Wallis test results show significant differences among the three groups (H = 12.5, *p* = 0.002). The H-value represents the magnitude of difference, while the *p*-value indicates the reliability of the difference. As can be seen from Figure 8, box plot and rank plot analysis results demonstrate clear separation among the three groups, supporting the findings from the Kruskal–Wallis test. Table 4 shows the results of Dunn’s post hoc analysis. The results indicate that the M group was significantly different from the C group (*p* = 0.001), while no significant differences were found between the M group and S Group, or between the S group and C group (*p* = 0.21). The above analysis demonstrates that the selection of scanning strategy is a determining factor for the fatigue life of components fabricated by SLM.

### 3.2. Analysis of Experiment Results Based on Microstructure

The fracture morphology of SLM 316L stainless steel specimens formed by different scanning strategies is shown in Figure 9. As can be seen from Figure 9, the surface of the fracture is uneven, with significant undulations, a dark gray color, and no metallic luster. The fracture surface is distributed with a large number of pits and holes, and all three types of specimens conform to the characteristics of ductile fracture. In the process of selective laser melting, due to the absorption of high-energy laser beam energy by 316L metal powder, the temperature rapidly rises, causing the metal powder to melt and solidify rapidly. Moreover, the forming process is influenced by factors such as material properties, process parameters, and equipment conditions, and defects are unavoidable. The defects generated during the SLM process are the most significant factor affecting fatigue performance, and their impact often exceeds the inherent fatigue strength of the material itself. The essence of fatigue failure is the process of crack initiation and propagation. For traditional forgings, crack initiation typically requires many cycles (accounting for over 90% of the fatigue life). However, for SLM components, internal defects act as pre-stored stress concentration points, greatly shortening or even completely eliminating the crack initiation stage, allowing the material to directly enter the crack propagation stage, resulting in a significant reduction in fatigue life.

As can be seen from Figure 9d–f, the specimens formed by the meander scanning strategy have the most defects, while the specimens formed by the chessboard scanning strategy have the fewest defects. The meander scanning strategy is easy to program and has high scanning efficiency, but the laser scans back and forth, and the directions of adjacent melt paths are opposite. This partially offsets residual stress, but strict control of interlayer cooling time is required. Sudden changes in scanning direction can lead to uneven energy input, which can easily form unfused pores or spatter defects in the molten pool at the reversal point. The stripe scanning strategy divides the processing layer into multiple parallel stripes, and bidirectional scanning is used within the stripes. The thermal distribution between adjacent stripes is optimized by rotating the scanning direction to avoid thermal stress concentration caused by continuous scanning of long paths. Its impact on defects is significantly different from meander scanning. If the stripe width is too small (<5 mm), increasing the number of overlapping areas can easily form pores at the boundary; if the stripe width is too large (>20 mm), it approaches meander scanning and thermal accumulation intensifies. In 316L stainless steel, stripe scanning (10 mm wide, 20% overlap) reduces porosity by 40% to 60% compared to meander scanning (from 1.2% to 0.5%). The chessboard scan strategy is a strategy that divides a single-layer processing area into multiple small, independent rectangular blocks (called “islands”) and scans them separately. Its core principle is to break down continuous long scan vectors and decompose the concentrated high thermal stress field into multiple small, independent, and randomly distributed thermal stress fields, thereby suppressing defects related to thermal stress at the root. The main risk of the chessboard strategy lies in the boundaries between islands. If the scanning parameters (power, speed, overlap rate) for boundary overlap are not set properly, or if there is a small positioning error during laser jumping, it is extremely easy to form small incomplete fusion defects at the boundary. The chessboard scanning strategy is a powerful double-edged sword. It performs well in controlling stress, deformation, and cracks, but at the cost of introducing a new defect risk of “island boundary non-fusion” and placing higher demands on equipment positioning accuracy. The key to successfully applying this strategy lies in finely optimizing the boundary parameters and overlap rate, in order to leverage strengths and avoid weaknesses, and obtain components with excellent comprehensive performance.

Figure 10 shows the crystal morphology photos and grain size distribution diagrams of the specimens formed by meander scanning, stripe scanning, and chessboard scanning, respectively. SLM formed specimens are mainly composed of equiaxed crystals, columnar crystals, and cellular crystals. The columnar crystals formed by the chessboard scanning strategy are smaller and denser than those formed by the other two scanning strategies. Compared with the specimens formed by the other two scanning strategies, the equiaxed crystal area of the specimens formed by the chessboard scanning strategy is larger, indicating that under the same processing conditions, the comprehensive mechanical properties of the specimens formed by the chessboard scanning strategy are the best. Columnar crystals grow preferentially along the fusion line perpendicular or at a certain angle to the fusion line. Analysis suggests that their crystallization mode is non-uniform nucleation growth based on the fusion zone, and the gaps between powders affect their thermal conductivity. Therefore, the heat in the melt pool mainly diffuses through the substrate and solidified parts towards the substrate, with a large temperature gradient in the direction of heat flow, which is more conducive to the growth of columnar crystals. During the layer-by-layer scanning process, due to changes in solute distribution in the alloy powder, the melting point of the liquid phase is altered, resulting in significant undercooling in the direction of layer accumulation in the formed specimen. Columnar crystals grow along the direction perpendicular to the fusion line. The micro melt pool is affected by the Gaussian laser mode and has a surface tension gradient. Under the action of surface tension gradient, thermal convection occurs in the molten pool. During this process, the flowing molten flow not only causes deformation of the molten pool, but also changes the direction of heat dissipation, resulting in preferential growth of columnar crystals along a certain angle. During the SLM specimen forming process, the different cooling rates of the melted metal powder result in varying grain sizes, as shown in Figure 10b. The fusion line between two bonding layers melted by selective laser shows an arc shape. Under the influence of Gaussian laser mode, the energy in the middle is the highest, and the energy at both ends gradually decreases. Analysis shows that the depth of the melting zone is different due to the difference in laser energy, resulting in an arc shape between the two bonding layers.

The crystal morphology core formed by SLM meander scanning and stripe strategies consists of coarse columnar crystals grown epitaxially along the construction direction, sub-micron-level cellular substructures, and significant anisotropy. This morphology is due to the highly oriented heat flow and extremely fast cooling rate during the scanning process. Directional growth of columnar crystals can lead to anisotropy in the mechanical properties of formed specimens. Usually, the performance in the vertical construction direction (XY plane), such as tensile strength and ductility, is superior to that in the parallel construction direction (Z direction). This is because when subjected to force in the Z direction, the load is more easily transmitted along vertical columnar grain boundaries, which are relatively fragile. Meanwhile, areas with poor fusion are also more likely to become pathways for crack propagation. The chessboard strategy is usually combined with interlayer rotation (such as rotating each layer by 67°). This can further disrupt the direction of heat flow, prevent excessive growth of grains along the construction direction, and make grain orientation more randomized. Splitting the large-scale scan into multiple small islands for scanning avoids severe heat accumulation caused by long scan lines, allowing each small island to cool down quickly. A faster cooling rate typically means a higher temperature gradient (G) and solidification rate (R), which facilitates the formation of finer grains. The change in G/R ratio also affects the grain morphology (the ratio of columnar crystals to equiaxed crystals). In summary, the SLM chessboard scanning strategy can effectively refine grains, weaken texture, and reduce anisotropy by segmenting the thermal field, randomizing the scanning sequence, and coordinating interlayer rotation. The improvement of these microstructures usually means better overall mechanical properties and lower crack sensitivity.

### 3.3. Analysis of Experiment Results Based on Residual Stress

As shown in Figure 11, SLM 316L stainless steel specimen 1, specimen 2, and specimen 3 were manufactured by the meander scanning, stripe scanning and chessboard scanning strategies, respectively. Nine measurement points are evenly distributed along the centerline of the x-direction, with a spacing of 15 mm between each measurement point. The blind hole method was used for the stress detection of SLM metal formed specimens. The basic idea of the blind hole method is to drill a small blind hole on the surface of a component with a certain initial stress, and then release some stress on the surface of the blind hole to produce corresponding displacement and strain. The stress magnitude used is calculated based on the deformation around the hole. The testing equipment included HK21A (GR/X-350A, Jinan, China). In order to ensure a tight bond between the strain gauges and the surface of the tested specimen, it is necessary to treat the SLM metal formed specimens and wipe their surfaces clean with alcohol. The measurement points locations (Labeled as 1, 2, 3, 4, 5, 6, 7, 8, 9) are shown in Figure 10. The diameter of the hole was 1.8 mm, and the depth was 0.8 mm. Table 5 shows the stress testing results of blind hole method. The residual stress distributions under different scanning strategies are shown in Figure 12. Figure 13 shows a comparison of residual stresses under different scanning strategies.

As shown in Figure 12, the stress ***σ_x_*** along the X-direction and the stress ***σ_y_*** i along the Y-direction both exhibit a macroscopic distribution characteristic of “tension compression alternation”. For the specimen formed by meander scanning, there is compressive stress at the edge of the specimen; as the measurement point moves away from the edge, the compressive stress gradually transforms into tensile stress. There is a difference in stress at the center of the specimen, and the tensile stress gradually increases towards the center, with a maximum value of 123.2 MPa. Because the shrinkage of the material is continuous and cumulative in the scanning direction, each newly solidified metal attempts to contract with the previously solidified metal, resulting in the continuous accumulation and enhancement of stress in the scanning direction. In the direction perpendicular to the scanning direction, shrinkage mainly occurs within a single pass or between adjacent passes, with relatively weak constraints and insignificant stress accumulation effects. The alternating rotation scanning angle (rotating 67° layer by layer) and remelting strategy can effectively disperse the directionality of stress and reduce the overall stress level.

Stripe scanning is an optimized variant of meander scanning, which divides a large scanning area into multiple narrow stripes and uses alternating scanning directions to concentrate the distribution of heat dissipation, effectively controlling residual stress. As can be seen from Figure 13, compared with meander scanning, the overall stress level is significantly reduced, and the maximum stress value is −45.3 MPa. The reason is that each stripe is a relatively independent thermal cycling unit during scanning. The boundaries between the stripes (unmelted powder or slightly remelted areas) act as a “stress isolation wall”, preventing the continuous transmission and accumulation of stress and deformation throughout the entire layer, effectively reducing stress levels. As can be seen from Figure 12c,d, due to the opposite scanning directions of adjacent stripes, the macroscopic stress directionality is greatly weakened. The scanning direction of a stripe is 0°, and the adjacent stripes are 180°. This means that the higher longitudinal tensile stress in one stripe will be partially offset by the same type of stress in the opposite direction in adjacent stripes. From the macroscopic scale of the entire component, material properties exhibit lower directional dependence (isotropic enhancement).

As shown in Figure 12e,f, for the specimen formed by the chessboard scanning strategy, the residual stress distribution is more uniform. Chessboard scanning divides a large area into small cells and frequently switches scanning positions, avoiding prolonged continuous scanning in the same direction, effectively dispersing the concentrated distribution of heat and reducing temperature gradients. This makes the residual stress distribution more uniform compared to long stripe scanning. Figure 12 shows a comparison of residual stress under different scanning strategies. For the specimen formed by chessboard scanning, the distributions of ***σ_x_*** and ***σ_y_*** are basically identical. The overall stress values were smaller than the specimen formed by the meander and stripe strategies.

In summary, the scanning strategy in SLM is an effective and commonly used process for regulating residual stress distribution. Different scanning strategies fundamentally affect the distribution and accumulation of heat by changing the path of heat source movement, thereby determining the generation, magnitude, and distribution of residual stress. The residual tensile stress in SLM parts has a significant negative impact on fatigue life, as it promotes crack initiation and propagation, and synergistically accelerates fatigue failure with other defects. Compressive residual stress is beneficial for fatigue life.

## 4. Conclusions

The research conclusions are based on the SLM specimen preparation conditions and experimental testing conditions provided in this study.

(1)The scanning direction determines the growth direction of the grains (epitaxial growth), thereby affecting the anisotropy and performance of SLM 316L stainless steel parts. During the SLM forming process, grains will grow epitaxially from existing crystals along the construction direction. Meander scanning can cause columnar grains to exhibit significant directionality within the scanning plane, leading to anisotropy in mechanical properties (including fatigue resistance). Stripe scanning and chessboard scanning continuously change the direction of heat flow, dispersing the directional growth of columnar crystals and promoting the formation of finer and more uniform equiaxed crystals, thereby improving the uniformity of the structure and overall fatigue performance.(2)Defects are the ‘origin’ of fatigue cracks, and the vast majority of fatigue cracks originate from surface or near-surface defects of SLM 316L stainless steel parts. Meander scanning is prone to forming continuous defects, while unidirectional long scanning lines can easily connect pores or incomplete fusion zones in the scanning direction, forming defect bands similar to pre-cracks, greatly reducing fatigue life. Partition scanning effectively isolates defects, while chessboard scanning and stripe scanning divide large areas into small units. Even if there are defects within a unit, these defects are confined within the unit, making it difficult to form a long-range continuous defect band, which is equivalent to interrupting the preferred path of fatigue cracks.(3)The scanning strategy affects the distribution and magnitude of residual stresses by changing the path of heat source movement. The meander scanning strategy can generate directional tensile residual stresses. Under cyclic loading, the superposition of working stress and residual tensile stress significantly increases the effective stress amplitude, enabling the SLM 316L stainless steel part to reach the fatigue limit faster and promoting crack initiation and propagation. Compressive stress is a beneficial ‘barrier’, and chessboard scanning can introduce compressive stress, counteract some of the working tensile stress, and even close cracks, thereby delaying the fatigue process and greatly improving the service life. Anisotropy leads to weak directions, while meander scanning results in poorer fatigue performance in SLM 316L stainless steel part parts in the scanning direction compared to other directions. Stripe scanning and chessboard scanning tend to make performance isotropic, without clear weak directions.

## Figures and Tables

**Figure 1 materials-18-04571-f001:**
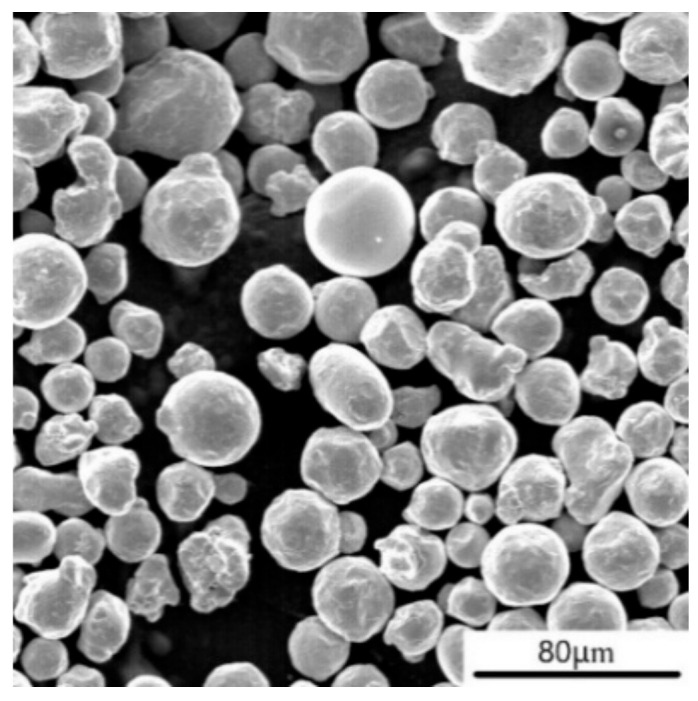
The SEM morphology of 316L.

**Figure 2 materials-18-04571-f002:**
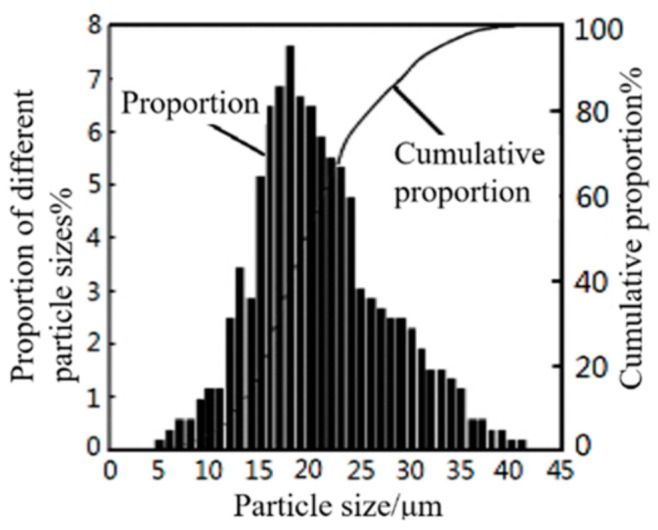
Particle size distribution diagram of 316L stainless steel powder of 316L stainless steel powder; d10 = 12.53 μm, d50 = 20.46 μm, d90 = 31.65 μm, measurement method: laser diffraction, sample size: 0.8 g.

**Figure 3 materials-18-04571-f003:**
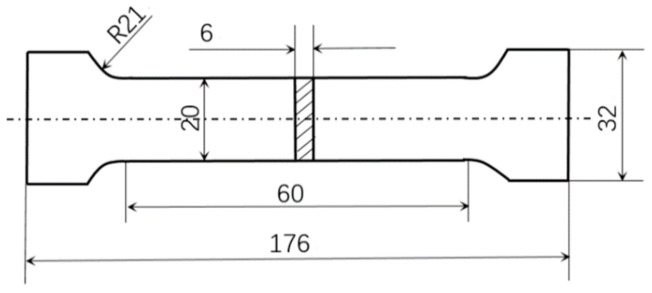
Schematic diagram of the specimen.

**Figure 4 materials-18-04571-f004:**
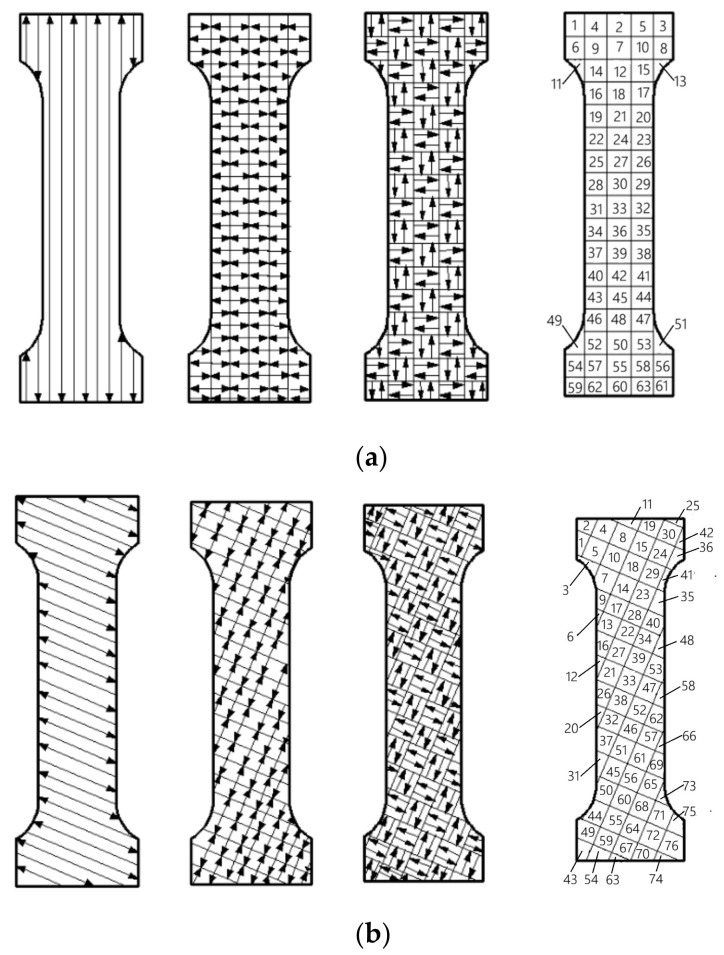
Schematic diagram of scanning path of meander scanning, stripe scanning, chessboard scanning, and the sequence of chess board scanning. (**a**) Scanning paths of layer N; (**b**) scanning paths of layer N + 1.

**Figure 5 materials-18-04571-f005:**
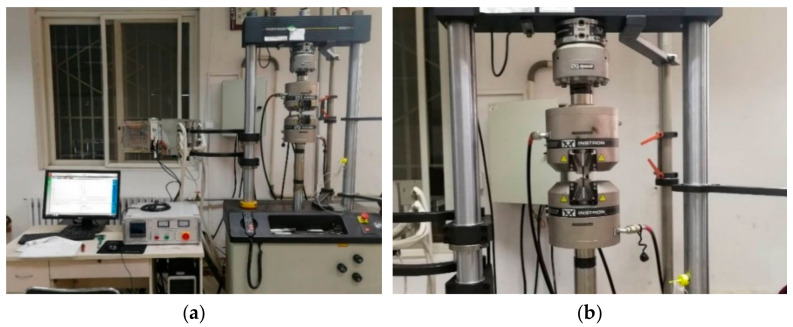
Fatigue testing equipment (**a**) Fatigue testing system; (**b**) Instron 8801 fatigue loading testing machine.

**Figure 6 materials-18-04571-f006:**
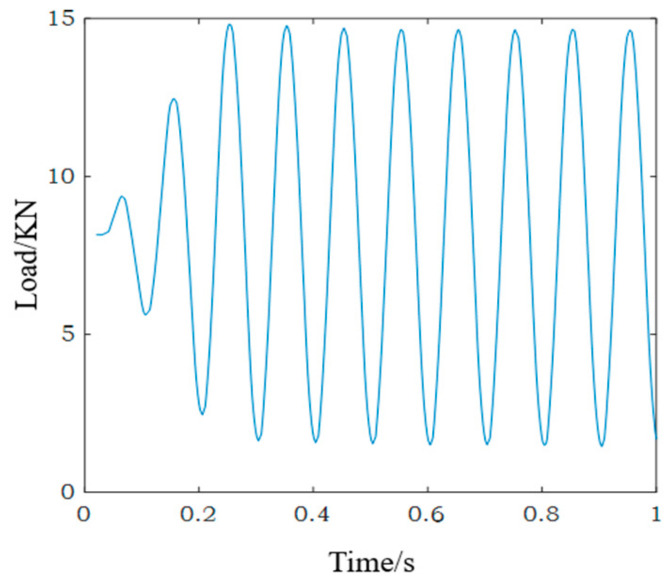
The fatigue load.

**Figure 7 materials-18-04571-f007:**
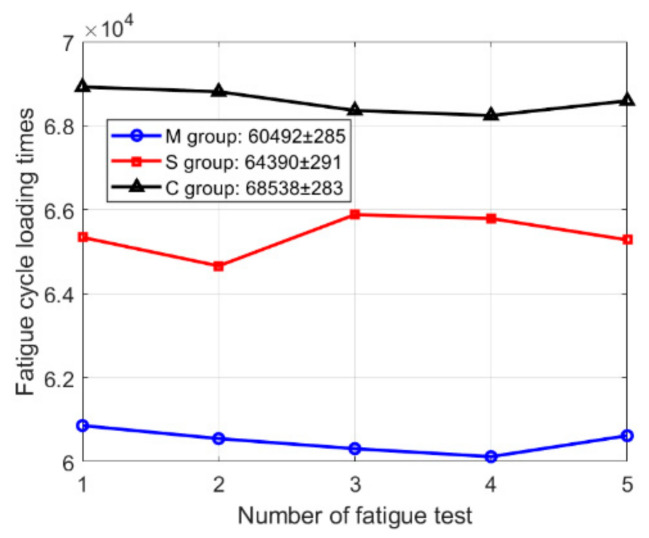
The fatigue loading results of SLM 316L stainless steel specimens formed by different scanning strategies.

**Figure 8 materials-18-04571-f008:**
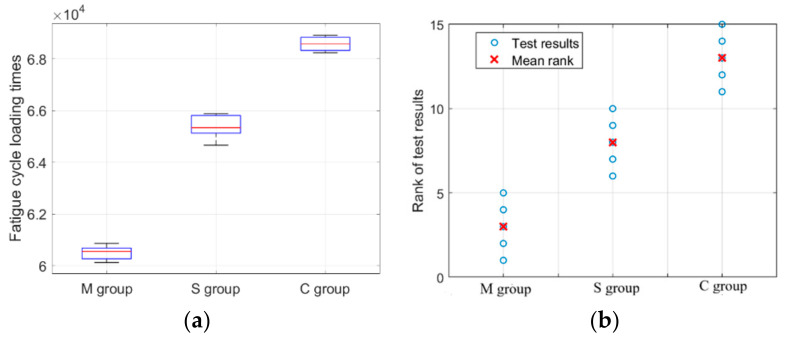
Results of box plot and rank plot analysis showing group distributions and rank patterns. (**a**) Box plot analysis results. (**b**) Rank plot analysis results.

**Figure 9 materials-18-04571-f009:**
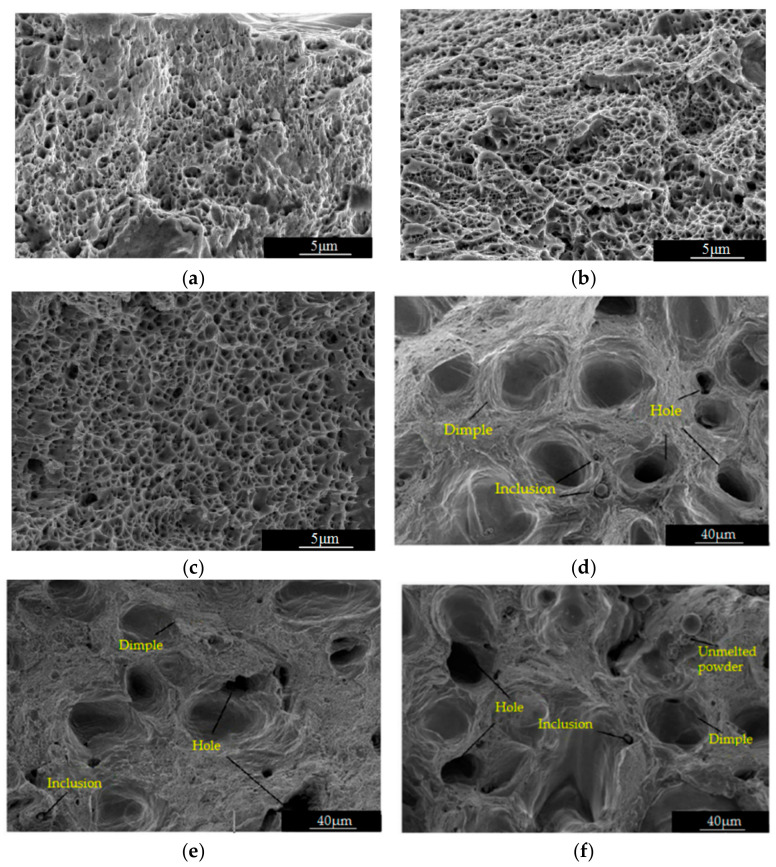
Fracture morphology of SLM 316L stainless steel specimens. (**a**) Specimen formed by meander scanning strategy, magnified 1000×. (**b**) Specimen formed by striped scanning strategy, magnified 1000×. (**c**) Specimen formed by chessboard scanning strategy, magnified 1000×. (**d**) Specimen formed by meander scanning strategy, magnified 10,000×. (**e**) Specimen formed by stripe scanning strategy, magnified 10,000×. (**f**) Specimen formed by chessboard scanning strategy, magnified 10,000×.

**Figure 10 materials-18-04571-f010:**
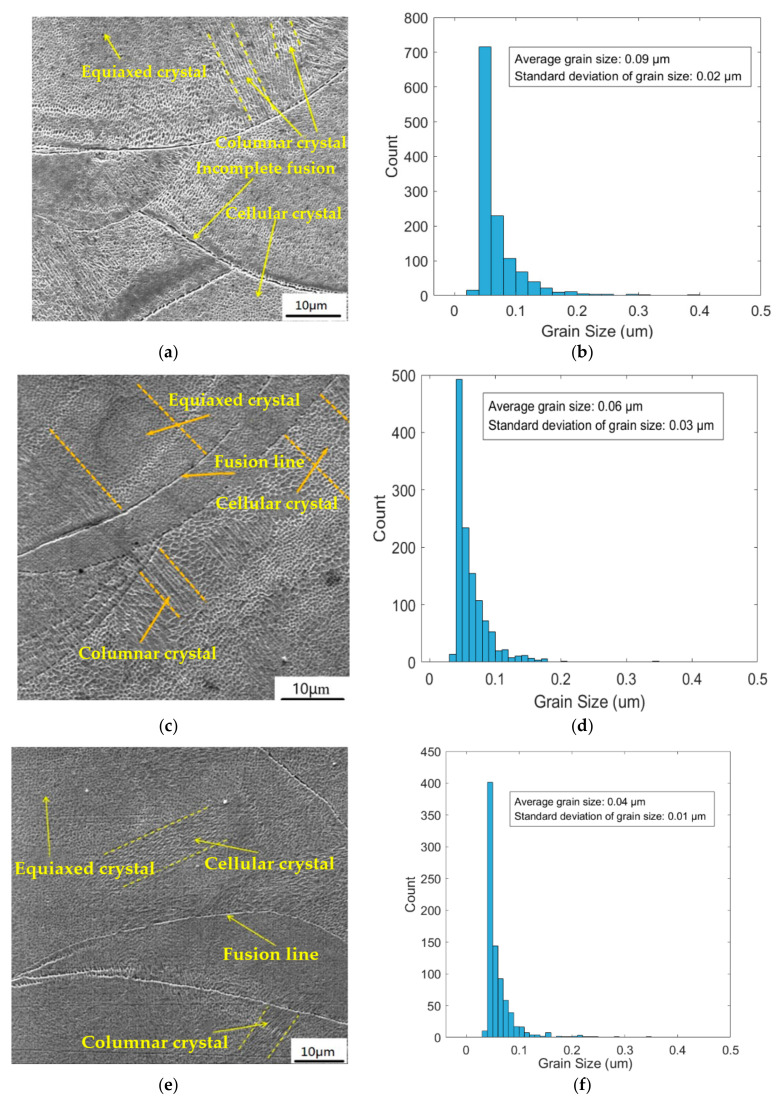
The crystal morphology of SLM 316L stainless steel specimens. (**a**) Meander scanning. (**b**) Grain size distribution diagram of meander scanning. (**c**) Stripe scanning. (**d**) Grain size distribution diagram of stripe scanning. (**e**) Chessboard scanning. (**f**) Grain size distribution diagram of chessboard scanning.

**Figure 11 materials-18-04571-f011:**
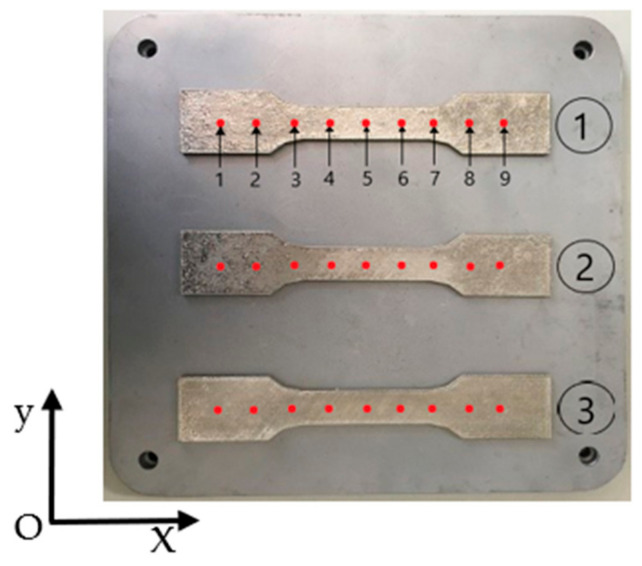
Schematic diagram of residual stress measurement points.

**Figure 12 materials-18-04571-f012:**
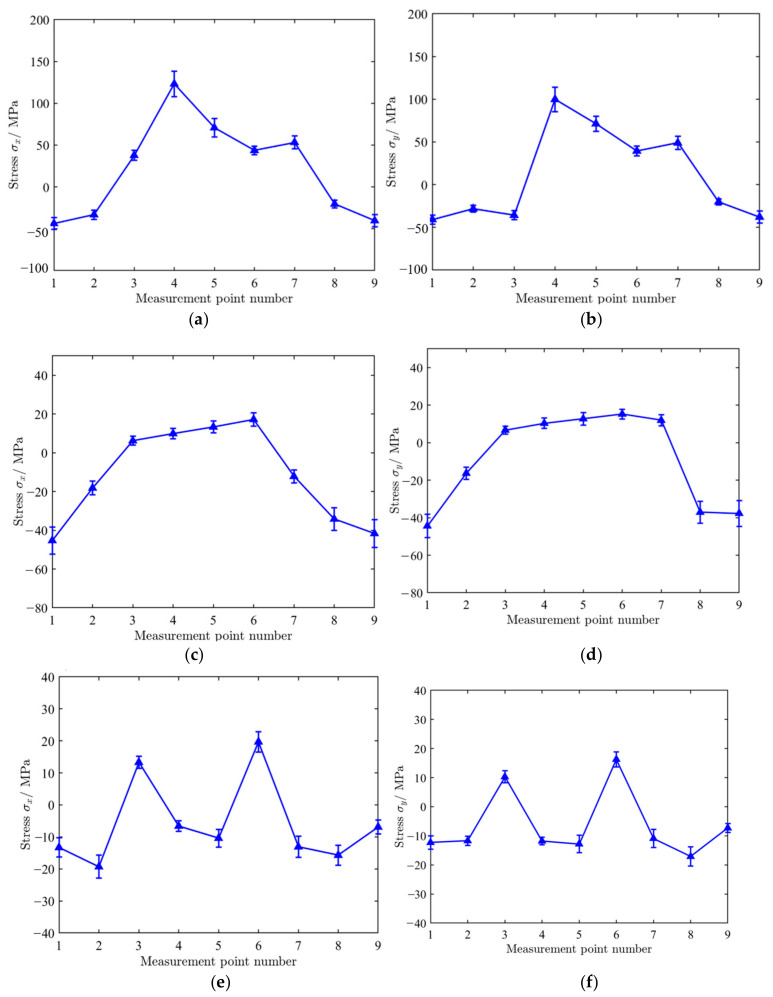
Residual stress distribution under different scanning strategies. (**a**) The distribution of***σ_x_*** −along the X-direction (meander scanning strategy); (**b**) the distribution of ***σ_y_*** along the Y-direction (meander scanning strategy); (**c**) the distribution of ***σ_x_*** along the X-direction (stripe scanning strategy); (**d**) the distribution of ***σ_y_*** along the Y-direction (stripe scanning strategy); (**e**) the distribution of ***σ_x_*** along the X-direction (chessboard scanning strategy); (**f**) the distribution of ***σ_y_*** along the Y-direction (chessboard scanning strategy).

**Figure 13 materials-18-04571-f013:**
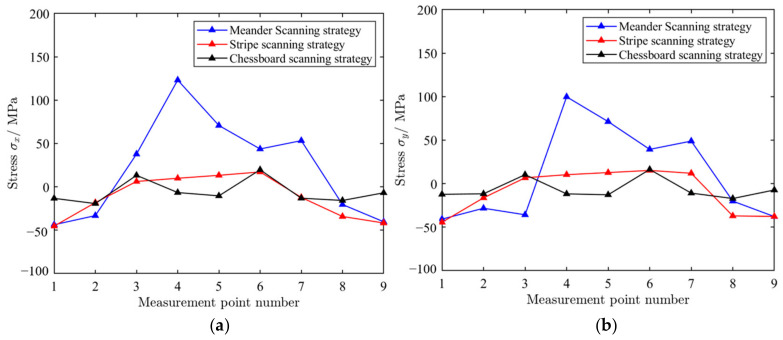
A comparison of residual stress under different scanning strategies. (**a**) The distribution of ***σ_x_*** along the X-direction. (**b**) The distribution of ***σ_y_*** along the Y-direction.

**Table 1 materials-18-04571-t001:** Chemical compositions of 316L stainless steel spherical powder (mass fraction, %).

Element	C	Cr	Ni	Mo	Si	Mn	O	*p*	Fe
Content	0.03	17.6	12.06	2.26	0.86	2	<0.1	0.04	Bal.

**Table 2 materials-18-04571-t002:** Main processing parameters.

Laser Power (W)	Scanning Speed (mm/s)	Layer Thickness (μm)	Scanning Interval (μm)	Spot Diameter (μm)	Volume Fraction of Oxygen (%)
300	750	30	60	70	≤0.03

**Table 3 materials-18-04571-t003:** Tensile properties of SLM specimens formed by different scanning strategies.

Specimen Number	Tensile Strength (MPa)	Yield Strength (MPa)	Elastic Modulus (N/mm^2^)	The Maximum Strain (%)
M-1	527	461	114,012	17.4
M-2	529	471	115,096	17.2
M-3	531	450	116,493	17.9
M-4	518	454	114,078	18.3
M-5	516	470	114,165	17.7
Mean	524.20	461.20	114,770	17.7
Standard deviation	6.76	9.36	1060.00	0.43
S-1	596.3	501	151,063	27.5
S-2	591.8	497	151,798	26.5
S-3	596.4	509	154,273	29.3
S-4	584.2	507	154,765	27.4
S-5	604	503	151,371	28.8
Mean	594.50	503.40	152,654	27.90
Standard deviation	7.25	4.77	1731.20	1.13
C-1	627	543	177,904	33.8
C-2	634.2	557	177,689	34.9
C-3	643.7	542	176,973	34.6
C-4	624.5	547	177,521	35.0
C-5	641.9	561	176,343	36.1
Mean	634.30	550.00	177,286	34.80
Standard deviation	8.59	8.54	629.84	0.82

**Table 4 materials-18-04571-t004:** Group comparisons.

Group Comparison	Difference	95% Confidence Interval	*p*-Value	Significance
M group vs. S group	−5.0	[−11.75, 1.75]	0.21	
M group vs. C group	−10.0	[−16.75, −3.24]	0.001	**Highly** **significant**
S group vs. C group	−5.0	[−11.75, 1.74]	0.21	

**Table 5 materials-18-04571-t005:** Stress testing results of blind hole method.

Measurement Point	*σ_x_*/MPa			*σ*_y_/MPa		
	Meander Scanning	Stripe Scanning	Chessboard Scanning	Meander Scanning	Stripe Scanning	Chessboard Scanning
1	−43.7	−45.3	−13.3	−40.8	−44.4	−12.3
2	−33.2	−18.2	−19.3	−28.3	−16.3	−11.7
3	37.8	6.2	13.3	−35.8	6.7	10.3
4	123.2	9.9	−6.6	99.8	10.3	−11.8
5	70.8	13.3	−10.4	71.3	12.7	−12.8
6	43.7	17.18	19.7	39.4	15.2	16.3
7	53.3	−12.2	−13.1	48.9	11.9	−10.9
8	−20.4	−34.2	−15.7	−20.1	−37.1	−17.1
9	−40.3	−41.7	−6.9	−37.9	−37.7	−7.3

## Data Availability

The original contributions presented in this study are included in the article. Further inquiries can be directed to the corresponding author.

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
