# Peer review of "The Influence Mechanism of a Scanning Strategy on the Fatigue Life of SLM 316L Stainless Steel Forming Parts"

_materials, 2025, doi:10.3390/ma18194571_

Round 1
Reviewer 1 Report
Comments and Suggestions for Authors
The paper is devoted to the study of the effect of scanning strategy in selective laser melting (SLM) on the fatigue strength of parts made of 316L stainless steel. The relevance of the study is determined by the widespread introduction of additive technologies in industry, where the reliability and durability of products directly depend on their fatigue characteristics. The authors showed that the choice of scanning trajectory has a decisive effect on the distribution of residual stresses, the formation of microstructure and defects, and therefore on the service life of parts. The experiments confirmed that the staggered scanning strategy provides the most favorable combination of properties, reduces anisotropy and increases fatigue resistance. The significance of the work lies in the fact that it not only reveals the mechanisms of defect and stress formation, but also offers practical approaches to improving the performance characteristics of products. The results of the study are of practical importance for optimizing process parameters and increasing the reliability of components manufactured by additive manufacturing methods.
The article needs some revision:
1. The authors criticize the "blind and extensive" approach of previous studies, but do not provide specific examples of such work. This weakens the argument. It would be useful to cite at least 2-3 studies where the technique was actually ineffective and to show how the present work fills this gap.
2. Powder characteristics are provided, but there is no information on its particle size distribution (e.g., average diameter and standard deviation). The maximum and minimum values ​​do not provide a complete picture of the distribution, which can significantly affect the formation of the structure during SLM.
3. The choice of only three scanning strategies (meander, stripe, checkerboard) seems somewhat limited. Other options are known in the literature (e.g., island scanning, spiral). It is necessary to explain why these three strategies were chosen and whether the results can be extrapolated to other options.
4. The authors claim that the staggered strategy gives better results due to the reduction of residual stresses. However, a quantitative comparison with other strategies in terms of stress level is not shown. It is desirable to add numerical values, and not only qualitative explanations.
5. The staggered strategy is shown to provide the highest number of cycles before failure. However, the number of fatigue test specimens is not specified. If only one specimen of each type is tested, this reduces the reliability. It is necessary to emphasize how many specimens were used and to indicate the dispersion of the results.
6. The figures with the microstructure demonstrate the difference in grain sizes. However, there is no quantitative assessment in the text (e.g., average grain size, distribution by types).
7. The authors note the uniformity of the stress distribution, but do not provide specific figures. The table contains the values, but it is advisable to emphasize them in the text and explain how exactly this affected the fatigue characteristics.
8. The first conclusion repeats the abstract and does not add new information. It would be better to structure the conclusions so that they directly answer the stated goal and show practical significance.
The English language of the article is generally understandable, but there are numerous stylistic and grammatical shortcomings. There are often literal translations from another language ("The specimen can be formed faster...", "This blind and extensive research method..."), which make the text unnatural. It is necessary to:
improve the coherence of sentences, replacing short phrases with smoother structures;
review the use of articles (for example, "the fatigue life" instead of "fatigue life" in some places);
correct the agreement of tenses, since sometimes the description of experiments is in the present tense, and sometimes in the past;
unify the terminology (for example, "meander scanning strategy" is sometimes referred to as "meander scanning", sometimes as "scanning path").
Author Response
Thank you very much for the time and effort that you have put into reviewing the previous version of the manuscript. Your suggestions have enabled me to improve my work greatly. Appended to this letter is my point-by-point response to the comments raised by you. The comments are reproduced and my responses are given directly afterward in a different color (red).
Should you have any questions, please contact me without hesitate.
Kind regards
Yan XiaoLing

Reviewer 2 Report
Comments and Suggestions for Authors
This manuscript addresses an important topic on the influence of scanning strategy on the fatigue life of SLM 316L stainless steel and provides useful experimental results linking mechanical performance, microstructure, defects, and residual stress. However, the current version requires significant improvements before it can be considered for publication. The research gap is not sufficiently established, the novelty is not highlighted clearly, and critical methodological details such as sample size, build orientation, and residual stress measurement are missing. The interpretation of results, particularly the claims about anisotropy reduction and the importance of scanning strategy, is overstated without stronger evidence such as EBSD analysis or statistical validation. In addition, several presentation issues including figure references, unit consistency, typos, and unclear phrasing reduce readability. Addressing these major and minor concerns will substantially strengthen the rigor, clarity, and overall impact of the study.
- The research gap is not clearly described. The Introduction does not convincingly establish what has not been studied before and why this work is necessary.
- The review of prior work on scanning strategies is too general. It lacks a deep comparison of how meander, stripe, and checkerboard strategies specifically affect residual stress, microstructure, and fatigue properties.
- The claimed key mechanism, residual stress influencing fatigue performance, is not adequately framed with existing literature. The link between residual stress and fatigue strength should be more comprehensively reviewed.
- The role of specimen preparation and possible stress release or redistribution during part removal is not acknowledged in the context of residual stress studies.
- The number of specimens tested for fatigue and tensile experiments is not specified. Without sample size and scatter data, the statistical significance of results is uncertain.
- Details of build orientation, surface condition, and post-processing are missing. These factors strongly affect fatigue life and must be clarified.
- Residual stress measurement by the blind-hole method is insufficiently described. Assumptions, gauge configuration, and measurement locations should be explained.
- Fatigue results are presented as single values without scatter. Multiple tests and error bars are essential to validate the observed differences.
- The claim that checkerboard scanning reduces anisotropy is not well supported. SEM images alone cannot fully demonstrate this, and EBSD data would be required.
- Porosity levels are discussed with percentages, but the method of measurement is not described. It is unclear whether these values are from this study or literature.
- The connection between residual stress profiles and fatigue life is mentioned but not quantitatively analyzed. The interpretation remains qualitative.
- The statement that scanning strategy is the most important factor is too strong. The study only varied scanning strategy, so its relative importance compared to other process parameters cannot be claimed.
- The absence of EBSD analysis weakens the conclusion about anisotropy reduction and equiaxed grain formation. SEM alone is insufficient.
- The Introduction should more clearly highlight the novelty of combining fatigue testing, residual stress measurement, and microstructural analysis under different scan strategies.
- Terminology and phrasing should be more precise. The current text sometimes reads as a general description rather than a critical review.
- The chosen fatigue loading condition of 400 MPa, R = 0.1, 10 Hz should be justified as representative or compared against material properties.
- The rationale for fixing process parameters such as laser power and scan speed while only varying scanning strategy should be explained.
- Figure and table numbering inconsistencies should be corrected.
- Fractography images would benefit from clearer annotation of crack initiation sites.
- The residual stress results are important but should be discussed with consideration of measurement uncertainty and part removal effects.
- Some long sentences and repetitive descriptions reduce clarity. Tighter writing would strengthen the discussion.
- The conclusions should be phrased more cautiously, highlighting findings within the limits of the present study such as single stress level and fixed process parameters.
- Units are not formatted consistently. For example, “400MPa” should be written as “400 MPa”.
- There are frequent typos and spacing errors throughout the manuscript, which should be carefully corrected for professionalism and readability.
Author Response

(The authors gave the same response as above.)

Round 2
Reviewer 1 Report
Comments and Suggestions for Authors
The work has been significantly improved: literary counterexamples have been provided, the choice of strategies has been clarified, quantitative data on residual stresses has been added, and the physical interpretation has been strengthened. However, to fully address the comments, the following are critical: adding numerical parameters for powder distribution, providing comprehensive fatigue test statistics (n, mean, SD/CI, significance tests), and introducing at least one quantitative grain size metric. Additionally, terminology and typos should be cleaned up.
1. Powder particle distribution – add numbers.
Add a table/row to the text with d10, d50, d90 (or mean ±σ), measurement method (laser diffraction, etc.), and sample size. Currently, there is only a "diagram." This directly addresses point 2.
2. Explicitly indicate n for fatigue for each strategy (if 5, show all 5 results). Provide mean ± SD/SE and confidence intervals; box plots or data points are preferred. Significance testing of differences (ANOVA/Kruskal-Wallis with post-hoc analysis) would strengthen the conclusions.
3. Even if the primary focus is on mechanisms, at least one metric (mean equiaxic/columnar grain size according to ASTM E112 or equivalent) would significantly strengthen the argument and eliminate point 6. A representative sample and a brief table may suffice.
4. The text contains inconsistencies and typos: "Meader"/"Meander," "Reinshaw" instead of "Renishaw," confusion between "checkerboard" and "chessboard," "loose density" instead of the generally accepted "apparent density," and minor grammatical errors.
Author Response
Dear Reviewer:
Thank you very much for the time and effort that you have put into reviewing the previous version of the manuscript. Your suggestions have enabled me to improve my work greatly. Appended to this letter is my point-by-point response to the comments raised by you. The comments are reproduced and my responses are given directly afterward in a different color (red).
Should you have any questions, please contact me without hesitate.
Kind regards
Yan XiaoLing

Reviewer 2 Report
Comments and Suggestions for Authors
The authors have adequately addressed the reviewer’s comments, and the manuscript has been improved significantly. I recommend acceptance in its current form.
Author Response

(The authors gave the same response as above.)
